# Defining healthcare never events to effect system change: A protocol for systematic review

**Joanna Zaslow**[1]*, **Jacqueline Fortier**[1], **Cara Bowman**[1], **Ria de Gorter**[1], **Ellen Tsai**[1], **Dennis Desai**[1], **Peter O'Neill**[1], **Richard Mimeault**[1], **Gary Garber**[1,2,3,4,5]

**1** The Canadian Medical Protective Association, Ottawa, ON, Canada, **2** Department of Medicine, University of Ottawa, Ottawa, ON, Canada, **3** School of Epidemiology and Public Health, University of Ottawa, Ottawa, ON, Canada, **4** Ottawa Hospital Research Institute, Ottawa, ON, Canada, **5** Department of Medicine, University of Toronto, Toronto, ON Canada

* Jzaslow@cmpa.org

## Abstract

### Introduction

A never event is the most egregious of patient safety incidents. It refers to events that should theoretically never happen, such amputating the wrong limb. The term "never event" is used around the world by a variety of medical and patient safety organizations and is synonymous with sentinel events and serious reportable events. Unfortunately, there is little consensus about which events, in particular, are never events. These differing lists hinder potential collaboration or large-scale analyses. A recent systematic review by Hegarty et al. (2020) identified the need for a standardized definition for serious reportable events. The objective of our systematic review is to build on this by identifying which events are consistently or frequently identified as never events in order to isolate those which are core never events.

### Materials and methods

A systematic review will be conducted using Medline, Medline in Process, Scopus, PsychINFO, Embase via OVID, and CINAHL via EBSCO databases, as well as grey literature. We will include articles of any study design that discuss never events or one of its synonymous terms in the context of medical care. Four independent reviewers will conduct the title and abstract as well as the full-text screening, and 2 reviewers will abstract data. Data will be analyzed using narrative synthesis. Results will be categorized by year and geographic location, and by other factors determined during full-text screening.

### Discussion and conclusion

The lack of consensus regarding never events hinders progress in reducing their occurrence. Differing data sources makes comparison challenging, and limits the ability for patient safety groups to work collaboratively and share learnings with others. Identifying a core set of never events will serve as a first step to focus our efforts to reduce these harmful incidents.

**Data Availability Statement:** No datasets were generated or analysed during the current study. All relevant data from this study will be made available upon study completion.

**Funding:** The authors received no specific funding for this work.

**Competing interests:** The authors have declared that no competing interests exist.

# Background

Healthcare-related harms, also known as adverse events, can encompass a wide variety of incidents, some of which are preventable [1]. The frequency of adverse events was highlighted in the Institute of Medicine's landmark report *To Err is Human*. This report emphasized how the combination of technological challenges, patient complexity, overworked personnel, and burnout created a healthcare system that faces constant stress, which can result in unanticipated errors [2]. While this report was released more than two decades ago, adverse events continue to occur worldwide. A recent systematic review calculated that 1 in 20 patients are exposed to preventable harm globally [3]. In Canada, research has estimated that patients experience harmful events in 1 out of 18 hospitalizations in Canada, and this rate has remained steady for several years [4]. In the United Kingdom it has been estimated that 1 in 20 hospitalizations involves an error, of which 6% are serious, [5] and in the United States, it is estimated that between 44,000–98,000 hospitalized patients die each year as a result of preventable medical errors [6].

"Never events" refers to a particular subset of adverse events that are considered the most egregious of healthcare-related errors, such as performing surgery on the wrong side or administering the wrong blood type in a transfusion. The term, first used by former CEO of the National Quality Forum Dr. Ken Kizer, [7] has been adopted by other healthcare organizations such as the US Centers for Medicare and Medicaid Services, [8] the National Health Service in the United Kingdom, [9] and the Canadian Patient Safety Institute [10]. The term itself is not without controversy; while some interpret it as a call to action to reduce adverse events, [10] others may equate its use as an assignment of responsibility or liability [11]. This may result in some providers hesitating to perform interventions on high-risk patients [12, 13] or to hiding incidents in order to lower event reporting data [14].

While there is much discussion about the risk and incidence of never events, there is little consensus on what the term means or what should be included under the category of a never event. Since the term was first introduced, patient safety researchers have applied several definitions with differing results. These definitions often include a combination of characteristics, such as that never events should be: (largely) preventable, unambiguous, and/or their prevention is important for public knowledge/confidence [7]. Due to the use of different definitions, organizations may not even classify the same incidents as never events. For example, there is disagreement as to whether wrong site surgery, catheter-assisted urinary tract infection, or patient death or injury due to medication errors are never events [7]. The National Quality Forum [NQF] lists death or serious injury associated with electric shock [15] as a serious reportable event, also known as a never event, along with several others that mention harm from physical restraints or bedrails, and oxygen or other gases. Of the events on that list, the Canadian Patient Safety Institute [CPSI] only includes harm related to "the administration of the wrong inhalation or insufflation gas" as a never event [10]. In total, the NQF never events list identifies 29 events, while CPSI identifies 16. Some have suggested that in broadening the lists of never events, there is a risk of diluting the overall efforts to reduce never events [16]. Furthermore, by including rare events in never events lists there will be less focus or fewer expended resources to reduce more common severe adverse events [16].

A recent systematic review of definitions of serious reportable patient safety incidents conducted by Hegarty et al. identified the need for a "standardized and consistent approach to defining serious incidence and associated reporting protocols," and proposed five dimensions of what they call serious reportable events [17]. We suggest that an important addition to this work is to identify which adverse events are consistently or frequently identified as never events. Therefore the aim of this systematic review is to answer the question: which patient

safety events are most frequently classified as never events? We will achieve this by performing a narrative synthesis of both peer-reviewed and grey literature articles that describe never events, either individually or as part of a framework, which encompasses a list of never events along with a unique definition and purpose for tracking never events. In performing this review, we aim to identify which never events are consistently reported, and thus can be seen as a core list for targeting system improvement.

## Methods/Design

### Review format

The systematic review will be carried out and reported according to the PRISMA 2020 guidelines [18].

### Definitions

Our definition of never events will rely on Hegarty et al.'s systematic review which generated a list of dimensions to define serious patient safety events: preventable; identifiable and measurable; run the risk of reoccurrence; cause unexpected or avoidable death or injury or have the potential to cause serious harm; and have the potential for learning [17]. Our work will also include events that are similarly labelled (e.g. serious reportable event), as long as they are consistent with the above definition.

### Inclusion and exclusion criteria

We utilized the SPIDER framework to support the development of the inclusion criteria and search strategy. This framework was developed for systematic reviews that utilize qualitative meta-syntheses [19].

### Information sources

An information specialist will search Medline, Medline in Process, and Scopus along with PsycINFO, Embase via OVID and CINAHL via EBSCO, during the electronic component of the scoping review. All databases will be searched from 2001 onward, the year "never event" was coined. There will be no language restrictions, or any other publication restrictions at the search stage.

### Search strategy

An information specialist will develop an electronic search strategy after consulting with the study team and identifying relevant MeSH terms and key terms. The search strategy will be peer reviewed using the PRESS checklist [20]. Both published and unpublished sources (i.e., grey literature) will be eligible for inclusion.

Additional sources will be identified using forward and backward searches of citations, drawing on article bibliographies for backward searches and Google Scholar's "Cited By" feature for forward searches.

## Study records

### Data management

The final search yield will be combined with articles derived from early literature reviews and stakeholder consultations and then undergo a removal of duplicates. All citations will be uploaded to Distiller SR (Evidence Partners, Ottawa, Canada).

### Selection process

Article selection will occur in two phases: an initial review of title and abstract, and a subsequent review of full article text. During both phases, reviewers will independently apply the eligibility criteria.

The reviewers will complete an initial calibration exercise to ensure consistency in the application of eligibility criteria. Inter-rater agreement will be quantified and assessed for an initial sample of 50 articles using a Kappa statistic in Distiller SR. The reviewers will repeat the calibration exercise until a target score of at least 0.61, the lower threshold for moderate agreement, [21] is achieved.

Following the completion of the calibration exercise, reviewers will independently review articles. Each article will be reviewed by two reviewers to determine eligibility. Discrepancies between reviewers will be resolved by consensus, with a third reviewer adjudicating if needed.

After full text review, eligible articles will be assessed for their methodological quality using the GRADE framework.

### Data collection

After completion of the selection process, data will be abstracted within Distiller SR using a data abstraction form. One reviewer will abstract the data from each article, and another reviewer will check the abstracted data for accuracy.

### Data items

The project team will determine which data items will be collected from included studies during data abstraction. See Table 1 for preliminary data collection items.

### Data analysis and synthesis

Data analysis will employ narrative synthesis and evidence mapping methods.

## Discussion

The term never events evokes a strong reaction, [7] as its associated events are those that are so egregious that they should never happen. Nevertheless, the data show that they continue to occur. Efforts to identify, report and ultimately prevent them are hindered, in part, by a lack of

**Table 1.**

| SPIDER Framework | Inclusion Criteria: | Exclusion Criteria: |
|---|---|---|
| **(S) S**ample | Articles describing patients interacting with the medical system | Any articles or reports that feature patient safety incidents that take place outside of the medical system (e.g. dentistry); or articles published before 2001 (when the term was first introduced) |
| **(PI) P**henomenon of **I**nterest | Patient safety events that meet our definition of never events | Any articles or reports about patient safety events that do not specifically list which events are never events |
| **(D) D**esign | Any study design, including peer-reviewed papers, regional/organizational guidelines, regional/organizational policies, or regional/organizational reporting papers | |
| **(E) E**valuation | Any | None |
| **(R) R**esearch Type | Peer-reviewed published literature (including empirical studies, literature reviews, commentaries, letters to the editor, and reports, book chapters), and grey literature (including white papers and policy documents). | Press releases and other announcements |

standard definitions and concepts [7]. The purpose of this systematic review is to identify which adverse events are commonly and consistently categorized as never events. From this review, we will extract a core list of never events that others could use to focus future interventions and collaborative work, and thus increase the potential for "greater tangible clinical benefits" and positive patient outcomes [3].

Data reporting is essential to improving patient care, by allowing clinicians to learn from past mistakes and facilitating the investigation of incidents [22]. Never event data has already been used for hospital reimbursement, [13, 23] assessing internal quality improvement projects, and public reporting [7]. However, using this data can be challenging, as measurement is inexact and comparisons may be ineffective due to the lack of a singular consensus list of never events [7]. This limits the ability for patient safety researchers and others to apply their learnings in larger-scale or comparative projects. Identifying a core list of never events can be a step towards making a singular list, which will then help in developing clear data guidelines for reporting and tracking incidents. Our next steps may include collaboration with other patient safety groups and healthcare organizations, agreeing on definitions for the included events and identifying best practices to reduce their incidence. On a smaller scale, healthcare organizations can use this core list to focus their prevention efforts by implementing systemic efforts to reduce or eliminate the risk of never events.

Serious healthcare-related harm is a pervasive issue affecting patients and preventable providers around the world, and never events are an important subset of serious healthcare-related errors. While it is easy to agree that never events ought to be eliminated, achieving that goal is challenging. Our systematic review will benefit healthcare professionals, patient safety researchers, and quality improvement specialists by contributing to a common understanding of never events, and help to focus efforts in reducing such harms.

## Supporting information

**S1 Appendix. PRISMA-P checklist.**
(DOCX)

**S2 Appendix. Literature search strategy.**
(DOCX)

**S3 Appendix. Preliminary data collection form.**
(DOCX)

## Acknowledgments

We would like to thank the following individuals for their assistance in preparing this protocol and future study: Craig MacKie, Lindsey Sikora, and Rob Hudson.

## Author Contributions

**Conceptualization:** Joanna Zaslow, Gary Garber.

**Methodology:** Joanna Zaslow, Jacqueline Fortier, Cara Bowman, Gary Garber.

**Writing – original draft:** Joanna Zaslow, Jacqueline Fortier, Cara Bowman.

**Writing – review & editing:** Joanna Zaslow, Jacqueline Fortier, Cara Bowman, Ria de Gorter, Ellen Tsai, Dennis Desai, Peter O'Neill, Richard Mimeault, Gary Garber.

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
