## [Decision Letter · Decision Letter 0]

15 Aug 2022

PONE-D-21-38367Defining healthcare never events to effect system change: A protocol for systematic reviewPLOS ONE

Dear Dr. Zaslow,

Thank you for submitting your manuscript to PLOS ONE. After careful consideration, we feel that it has merit but does not fully meet PLOS ONE’s publication criteria as it currently stands. Therefore, we invite you to submit a revised version of the manuscript that addresses the points raised during the review process.

Please submit the revised manuscript in the light of reviewer's comment.

We look forward to receiving your revised manuscript.

Kind regards,

Alok Ranjan

Academic Editor

PLOS ONE

Journal Requirements:

2. We note you have included a table to which you do not refer in the text of your manuscript. Please ensure that you refer to Table 1 in your text; if accepted, production will need this reference to link the reader to the Table.

Reviewers' comments:

Reviewer's Responses to Questions

**Comments to the Author**

1. Does the manuscript provide a valid rationale for the proposed study, with clearly identified and justified research questions?

Reviewer #1: Yes

2. Is the protocol technically sound and planned in a manner that will lead to a meaningful outcome and allow testing the stated hypotheses?

Reviewer #1: Partly

3. Is the methodology feasible and described in sufficient detail to allow the work to be replicable?

Reviewer #1: No

4. Have the authors described where all data underlying the findings will be made available when the study is complete?

Reviewer #1: Yes

5. Is the manuscript presented in an intelligible fashion and written in standard English?

Reviewer #1: Yes

6. Review Comments to the Author

You may also provide optional suggestions and comments to authors that they might find helpful in planning their study.

Reviewer #1: Thank you for the opportunity to review this paper. The authors describe a protocol for a systematic review on finding the most common list of never events, through looking at the academic and grey literature. The introduction justifies the need for this study, which nicely compliments Hegarty et al.’s systematic review of definitions of serious reportable events. Below are a few points for consideration to improve the strength of your protocol:

• In the abstract you mention “to identify ‘core’ never events” – the addition of the term “core” gives the reader the sense that you are looking to focus in on a subset of the larger list of never events, so perhaps a clarification on this would be good.

• On line 102, you mention “We will achieve this by performing a narrative synthesis of both peer-reviewed and grey literature for never events frameworks in order to identify those events which are consistently..”. It would be helpful to describe what never events “frameworks” refer to in this case – are these lists?

• Is there a justification for excluding articles published before 2001?

• The design section of the SPIDER framework includes national guidelines, national policy or national reporting systems.

Would peer reviewed articles outside this criteria not be included in the study (i.e. single studies looking at a single never event)? If this is the case, it would be important to highlight within your objectives that your systematic is primarily focused on lists from these sources as it changes the scope of the review.

• Since this is a protocol, it would be helpful to have a preliminary data extraction sheet in order to further help the reader understand what kind of data you are hoping to gather.

• Within your discussion, you mention that “Identifying a core of never events can be a step towards making a singular list”. Perhaps a line or two on what would be the next steps (e.g. Delphi process?)

7. PLOS authors have the option to publish the peer review history of their article (what does this mean?). If published, this will include your full peer review and any attached files.

Reviewer #1: No

---

## [Author Response · Author response to Decision Letter 0]

23 Sep 2022

Response to Reviewers:

1.In the abstract you mention “to identify ‘core’ never events” – the addition of the term “core” gives the reader the sense that you are looking to focus in on a subset of the larger list of never events, so perhaps a clarification on this would be good.

We thank the reviewer for the comment and have added further detail into the abstract to clarify that our objective is to identify events that are consistently labelled as never events (lines 35-37).

2.On line 102, you mention “We will achieve this by performing a narrative synthesis of both peer-reviewed and grey literature for never events frameworks in order to identify those events which are consistently...” It would be helpful to describe what never events “frameworks” refer to in this case – are these lists? 

We have further clarified what our search strategy entailed (see lines 104-108). We have kept the word “framework” but have offered further explanation of what this entails – this means a list of never events along with a unique definition for never events and a unique purpose for tracking them. 

3.Is there a justification for excluding articles published before 2001?

We limited our search to articles published after 2001, as that is the year when the term was first introduced (see Table 1 and line 128).

4.The design section of the SPIDER framework includes national guidelines, national policy or national reporting systems? 

We have added further explanation in Table 1, where we have identified any study design (including peer-reviewed papers, regional/organizational guidelines, policies, or reporting papers) as meeting our inclusion criteria. 

5. Would peer reviewed articles outside this criteria not be included in the study (i.e. single studies looking at a single never event)? If this is the case, it would be important to highlight within your objectives that your systematic is primarily focused on lists from these sources as it changes the scope of the review.

We have added further explanation that we will be including single studies looking at a single never event (see line 102). 

6. Since this is a protocol, it would be helpful to have a preliminary data extraction sheet in order to further help the reader understand what kind of data you are hoping to gather.

We appreciate this suggestion and have attached a preliminary data extraction sheet to provide additional context to readers. 

7. Within your discussion, you mention that “Identifying a core of never events can be a step towards making a singular list”. Perhaps a line or two on what would be the next steps (e.g. Delphi process?)

We have added further detail about our next steps, including possible consultations with patient safety groups and health care organizations (see lines 191-193). We wish to thank the reviewer for their thoughtful comments and feedback.

---

## [Decision Letter · Decision Letter 1]

1 Dec 2022

Defining healthcare never events to effect system change: A protocol for systematic review

PONE-D-21-38367R1

Dear Dr. Zaslow,

We’re pleased to inform you that your manuscript has been judged scientifically suitable for publication and will be formally accepted for publication once it meets all outstanding technical requirements.

Kind regards,

Alok Ranjan

Academic Editor

PLOS ONE

Additional Editor Comments (optional):

Reviewers' comments:

Reviewer's Responses to Questions

**Comments to the Author**

1. Does the manuscript provide a valid rationale for the proposed study, with clearly identified and justified research questions?

Reviewer #1: Yes

2. Is the protocol technically sound and planned in a manner that will lead to a meaningful outcome and allow testing the stated hypotheses?

Reviewer #1: Yes

3. Is the methodology feasible and described in sufficient detail to allow the work to be replicable?

Reviewer #1: Yes

4. Have the authors described where all data underlying the findings will be made available when the study is complete?

Reviewer #1: Yes

5. Is the manuscript presented in an intelligible fashion and written in standard English?

Reviewer #1: Yes

6. Review Comments to the Author

You may also provide optional suggestions and comments to authors that they might find helpful in planning their study.

Reviewer #1: Thank you for addressing all the comments - the updated edits & data abstraction sheet make this protocol methodology sufficiently detailed. Good luck!

7. PLOS authors have the option to publish the peer review history of their article (what does this mean?). If published, this will include your full peer review and any attached files.

Reviewer #1: No

---

## [Editor Report · Acceptance letter]

6 Dec 2022

PONE-D-21-38367R1 

Defining healthcare never events to effect system change: A protocol for systematic review 

Dear Dr. Zaslow:

I'm pleased to inform you that your manuscript has been deemed suitable for publication in PLOS ONE. Congratulations! Your manuscript is now with our production department. 

Kind regards, 

on behalf of

Dr. Alok Ranjan 

Academic Editor

PLOS ONE